

# Entanglement of exact excited eigenstates of the Hubbard model in arbitrary dimension

Oskar Vafek[1,2], Nicolas Regnault[1,3] and B. Andrei Bernevig[1]

**1** Department of Physics, Princeton University, Princeton, New Jersey 08544, USA
**2** National High Magnetic Field Laboratory and Department of Physics,
Florida State University, Tallahasse, Florida 32306, USA
**3** Laboratoire Pierre Aigrain, Ecole Normale Supérieure-PSL Research University, CNRS,
Université Pierre et Marie Curie-Sorbonne Universités, Université Paris Diderot-Sorbonne
Paris Cité, 24 rue Lhomond, 75231 Paris Cedex 05, France

## Abstract

We compute exactly the von Neumann entanglement entropy of the eta-pairing states - a large set of exact excited eigenstates of the Hubbard Hamiltonian. For the singlet eta-pairing states the entropy scales with the logarithm of the spatial dimension of the (smaller) partition. For the eta-pairing states with finite spin magnetization density, the leading term can scale as the volume or as the area-times-log, depending on the momentum space occupation of the Fermions with flipped spins. We also compute the corrections to the leading scaling. In order to study the eigenstate thermalization hypothesis (ETH), we also compute the entanglement Rényi entropies of such states and compare them with the corresponding entropies of thermal density matrix in various ensembles. Such states, which we find violate strong ETH, may provide a useful platform for a detailed study of the time-dependence of the onset of thermalization due to perturbations which violate the total pseudospin conservation.


# 1   Introduction

The question of how equilibration and thermalization arise in isolated quantum (many-body) systems led to the eigenstate thermalization hypothesis (ETH) [1–4]. ETH states that in the thermodynamic limit, the eigenstate expectation value of a few-body operator in a typical eigenstate of a many-body Hamiltonian at energy $E$ is equal to the microcanonical average at the mean energy per volume $E/V$. The two main interpretations of the ETH, the weak versus strong ETH, state that *almost all* versus *all* the finite energy density eigenstates of a many-body Hamiltonian appear thermal to all local measurements [5].

The ETH also has fundamental implications on quantum information-inspired quantities that characterize the excited states. More specifically, the entanglement spectrum and the resulting entanglement entropy have long become powerful diagnostics of topological order, gapless or gapped nature of ground-states, and other properties [6]. One implication of the ETH is that thermal states have volume law entanglement as opposed to area-type entanglement entropy of the ground state and low-lying excited states of the system. The volume law entanglement is then thought to return to an area law entanglement when/if the many-body localization sets in [4].

Unfortunately, the paucity of exact results makes it difficult to test or demonstrate ETH and its consequences in generic, non-integrable, many-body models in more than one space dimension with realistic electron-electron interactions. Numerical studies are limited to the very small system sizes imposed by the exact diagonalization. Motivated by the fact that a class of exact excited eigenstates of the Hubbard model is known [7, 8], that the number of such states is a exponentially large in volume [9], and that their energy density differs from the ground state energy density by a finite amount, here we obtain the *closed form exact* expressions for the entanglement spectrum, the von Neumann entanglement and Rényi entropies of such states. The entanglement entropy for these states shows either a $\ln(V)$ law, or a $V$ (volume) law, or even an area-times-log law, depending on the number and the momentum space distribution of the flipped spins in the state. When their entropy is sub-extensive, such states therefore clearly violate strong ETH. Even when the entropy scales with $V$, the pref-actor is independent of the Hubbard $U$, and is not expected to correspond to the entropy in the microcanonical average, which should be a non-trivial function of $U$. Despite being in the middle of the full Hubbard spectrum, the pure *spin singlet* eta-pairing states, which show

ln($V$) entanglement, are simultaneously the ground-states and the most excited states in their specific quantum number sectors. Kantian et.al. proposed an interesting way to prepare the eta-pairing state with cold atoms in optical lattice [10]. If successfully implemented, our results make a concrete prediction about the reduced density matrix of a small subsystem, and the precise way that the remainder serves as a thermal bath.

The eta-pairing states with *flipped spins* are richer. They display either volume or area-times-log entanglement, depending on the momentum space occupation of the flipped Fermions. We find that even for the states with volume law entanglement, the entanglement Rényi entropies do not match those of the thermal density matrix in the canonical ensemble. They match the Rényi entropy of the thermal density matrix in a grand canonical ensemble, but with additional constraints on the quantum numbers of the states.

## 2 The model

We consider the Fermionic Hubbard model on a hypercubic lattice in *any dimension*. The Hamiltonian is $\hat{H} = \hat{T} + \hat{V}$ where

$$\hat{T} = t \sum_{\langle \mathbf{rr'} \rangle, \sigma} \left( \hat{c}_{\mathbf{r}\sigma}^{\dagger} \hat{c}_{\mathbf{r'}\sigma} + \hat{c}_{\mathbf{r'}\sigma}^{\dagger} \hat{c}_{\mathbf{r}\sigma} \right) - \mu \sum_{\mathbf{r}\sigma} \hat{c}_{\mathbf{r}\sigma}^{\dagger} \hat{c}_{\mathbf{r}\sigma}, \tag{1}$$

$$\hat{V} = U \sum_{\mathbf{r}} \hat{c}_{\mathbf{r}\uparrow}^{\dagger} \hat{c}_{\mathbf{r}\uparrow} \hat{c}_{\mathbf{r}\downarrow}^{\dagger} \hat{c}_{\mathbf{r}\downarrow}, \tag{2}$$

$\hat{c}_{\mathbf{r}\sigma}^{\dagger}$ is the Fermionic creation operator at a site $\mathbf{r}$ (belonging to the hypercubic lattice) and spin projection $\sigma = \uparrow$ or $\downarrow$. The total number of sites is $M$ and the first sum is over the nearest neighbor links.

The exact, $2N$-particle, spin-singlet, normalized, eta-pairing eigenstate [7] of $\hat{H}$ that we firstly focus on is

$$|\psi_N\rangle = C_N \left( \sum_{\mathbf{r}} e^{i\pi \cdot \mathbf{r}} \hat{c}_{\mathbf{r}\downarrow}^{\dagger} \hat{c}_{\mathbf{r}\uparrow}^{\dagger} \right)^N |0\rangle, \tag{3}$$

where $C_N = \sqrt{\frac{(M-N)!}{M!N!}}$, and $\pi = (\pi, \pi, \ldots, \pi)$. This follows readily from the commutator of $\hat{H}$ and $\sum_{\mathbf{r}} e^{i\pi \cdot \mathbf{r}} \hat{c}_{\mathbf{r}\downarrow}^{\dagger} \hat{c}_{\mathbf{r}\uparrow}^{\dagger}$; its energy is $E_{\psi_N} = (U - 2\mu)N$ [7,8]. As shown by C.N. Yang [7], this state is *not* the ground state of the Hubbard model for either $U \lessgtr 0$. At half filing, $\mu = U/2$ and the energy of this state vanishes. For repulsive $U$, the ground state at half filling is an anti-ferromagnetic insulator [11] with negative energy per particle (see e.g. [12]). For attractive $U$ the ground states are an s-wave superconductor and a charge density wave [13], also with negative energy per particle. At weak coupling ($U \ll t$) and near half filing, $|\psi_N\rangle$ sits near the middle of the energy spectrum. That is because the weakly perturbed filled Fermi sea with momenta centered near $\mathbf{k} = 0$ is near the bottom of the many-body band and with momenta centered near $\mathbf{k} = \pi$ is near its top. In the Appendix we introduce a generalization of the Hubbard model Eq[2] for which there exist similar eta-pairing eigenstates.

We partition the $M$ sites into a group $A$ with $M_A$ sites and a group $B$ with $M_B = M - M_A$ sites and compute the reduced density matrix $\hat{\rho}_A$ by tracing all the degrees of freedom in the group $B$. We then take the thermodynamic limit $N \to \infty$, $M \to \infty$ such that the boson filling $N/M \to \nu \sim O(1)$. After this limit, we then take $M_A \gg 1$. The system $A$ is therefore small compared to $B$ so that $B$ can serve as its bath, but still large enough to allow scaling of its entanglement entropy.

# 3 Reduced density matrix

We now sketch the derivation of the reduced density matrix.[1] We use integration over the contour $\mathscr{C}$ encircling the origin in the complex $z$-plane counterclockwise to re-write the eta-pairing state as:

$$|\psi_N\rangle = C_N N! \oint_{\mathscr{C}} \frac{dz}{2\pi i} \frac{1}{z^{N+1}} e^{z \sum_{\mathbf{r}} e^{i\pi\cdot\mathbf{r}} \hat{c}^\dagger_{\mathbf{r}\downarrow} \hat{c}^\dagger_{\mathbf{r}\uparrow}} |0\rangle. \tag{4}$$

Terms in the sum $\sum_{\mathbf{r}} e^{i\pi\cdot\mathbf{r}} \hat{c}^\dagger_{\mathbf{r}\downarrow} \hat{c}^\dagger_{\mathbf{r}\uparrow}$ commute, therefore we can write the exponential of the sum as the product of the exponentials. Moreover since $(\hat{c}^\dagger_{\mathbf{r}\downarrow} \hat{c}^\dagger_{\mathbf{r}\uparrow})^2 = 0$ we see that the operator part of Eq. 4 becomes $\prod_{\mathbf{r}} \left(1 + z e^{i\pi\cdot\mathbf{r}} \hat{c}^\dagger_{\mathbf{r}\downarrow} \hat{c}^\dagger_{\mathbf{r}\uparrow}\right)|0\rangle$. We then obtain:

$$\hat{\rho}_A = \text{Tr}_B \left(|\psi_N\rangle\langle\psi_N|\right)$$
$$= \frac{(M-N)!N!}{M!} \oint_{\mathscr{C}} \frac{dz_1}{2\pi i} \oint_{\mathscr{C}} \frac{dz_2}{2\pi i} \frac{(1+z_1 z_2)^{M_B}}{(z_1 z_2)^{N+1}} e^{z_1 \sum_{\mathbf{r}\in A} e^{i\pi\cdot\mathbf{r}} \hat{c}^\dagger_{\mathbf{r}\downarrow} \hat{c}^\dagger_{\mathbf{r}\uparrow}} |0_A\rangle\langle 0_A| e^{z_2 \sum_{\mathbf{r}\in A} e^{-i\pi\cdot\mathbf{r}} \hat{c}_{\mathbf{r}\uparrow} \hat{c}_{\mathbf{r}\downarrow}}, \tag{5}$$

$|0_A\rangle$ denotes the state with all sites in the region $A$ empty. Only the same powers of $z_1$ and $z_2$ survive the contour integration. Expanding $(1+z_1 z_2)^{M_B}$ using binomial expansion, performing the contour integration and eliminating the sum coming from the binomial expansion, gives the entanglement spectrum:

$$\hat{\rho}_A = \sum_{k=0}^{M_A} \lambda_k |k\rangle\langle k|; \quad \lambda_k = \frac{\begin{pmatrix} M_B \\ N-k \end{pmatrix} \begin{pmatrix} M_A \\ k \end{pmatrix}}{\begin{pmatrix} M \\ N \end{pmatrix}}. \tag{6}$$

We assumed $M_A < N$. The states $|k\rangle$ are orthonormal eta-pairing states of the $A$ side:

$$|k\rangle = \sqrt{\frac{(M_A-k)!}{M_A! k!}} \left(\sum_{\mathbf{r}\in A} e^{i\pi\cdot\mathbf{r}} \hat{c}^\dagger_{\mathbf{r}\downarrow} \hat{c}^\dagger_{\mathbf{r}\uparrow}\right)^k |0_A\rangle. \tag{7}$$

A Vandermonde convolution confirms that $\sum_{k=0}^{M_A} \lambda_k = 1$. Similar result for a ferromagnetic Heisenberg model appears in Ref. [14]. Ref. [15] also studies the $\eta$-pairing state, but uses a different normalization; an expression for $\lambda_{\mathbf{k}}$ in which $N$ appears only via $N/M$ is quoted in [16].

Eq. 6 shows that for each $k$, the eigenvalue of the density matrix is equal to the number of ways to simultaneously place $k$ pairs on $M_A$ sites and $N-k$ pairs on $M_B = M - M_A$ sites, divided by the number of ways to place $N$ pairs onto $M$ sites. The system is subject to the constraint on no double pair occupancy. In the thermodynamic limit, the largest number of configurations corresponds to the uniform particle density, i.e. $\lambda_k$ should be very sharply peaked about $k_m = M_A(N/M)$.

In the limit of interest, we can use the Stirling formula $n! \approx \sqrt{2\pi n} e^{n(\ln n - 1)}$ where $n$ is large. Then,

$$\lambda_k \approx \frac{1}{\sqrt{2\pi\kappa}} e^{-\frac{1}{2\kappa}(k-k_m)^2}, \tag{8}$$

---

[1]We provide detailed derivations of our results in the Appendix, including an extension to the "generalized" Hubbard model.

where $\kappa = \nu(1-\nu)M_A$. This form is valid as long as $\nu$ is not infinitesimally close to 0 or 1. Substituting the above Gaussian form into the von Neumann entanglement entropy $S_A = -\sum_{k=0}^{M_A} \lambda_k \ln \lambda_k$, and replacing the discrete sum over $k$ with an integral we obtain that $S_A$ scales as the *logarithm* [15, 16] of the number of sites in the region $A$:

$$S_A = \frac{1}{2}\left(1 + \ln\left[2\pi\nu(1-\nu)M_A\right]\right). \tag{9}$$

The small value of $S_A$ *seems* to be in a contradiction with the ETH motivated expectation that finite energy density excited states in the middle of the many-body spectrum should thermalize with the entanglement entropy scaling as the volume ($\sim M_A$). However, it is not, due to the existence of additional pseudospin symmetry operators [8], and the eta-pairing states are the *only* states in their symmetry sector. The existence of a global conserved pseudospin [7,8] is special to the Hubbard model, and the corresponding operator is [7,8]

$$\hat{J}^2 = \frac{1}{2}\left(\hat{J}_+\hat{J}_- + \hat{J}_-\hat{J}_+\right) + \hat{J}_0^2, \tag{10}$$

where $\hat{J}_+ = \sum_{\mathbf{r}} e^{i\pi\cdot\mathbf{r}}\hat{c}^\dagger_{\mathbf{r}\downarrow}\hat{c}^\dagger_{\mathbf{r}\uparrow}$, $\hat{J}_- = \hat{J}_+^\dagger$, and $\hat{J}_0 = \frac{1}{2}(\hat{N}-M)$. Because $\hat{J}^2$ commutes with $\hat{J}_+$, the state $|\psi_N\rangle$ corresponds to the maximal eigenvalue of the $\hat{J}^2$, namely $\frac{M}{2}\left(\frac{M}{2}+1\right)$, independent of $N$. Different members of this highest $J = \frac{M}{2}$ multiplet have a different value of $J_0$ (hence different particle number). Note that the spin singlet pairs in $|\psi_N\rangle$ are not severed by the $A-B$ partition. If each state in this multiplet was equally likely, the entropy within this sector would be the logarithm of the multiplicity of the multiplet. There are $\sim M_A$ states in the multiplet which are accessible in the region with $M_A$ sites, hence $S_A \sim \ln M_A$. The pre-factor $\frac{1}{2}$ originates from Eq. 8 being Gaussian distributed with the width $\sim \sqrt{M_A}$.

The eta-pairing states have a natural generalization when $\hat{J}_+$ acts on *any* fully polarized states instead of the vacuum. This class of spin-flip eta-pairing states is:

$$|\psi_N^{\{\mathbf{k}\}}\rangle = \hat{c}_N^{\mathbf{k}}\left(\sum_{\mathbf{r}} e^{i\pi\cdot\mathbf{r}}\hat{c}^\dagger_{\mathbf{r}\downarrow}\hat{c}^\dagger_{\mathbf{r}\uparrow}\right)^N \prod_{\mathbf{k}\in\mathscr{F}} \hat{c}^\dagger_\downarrow(\mathbf{k})|0\rangle, \tag{11}$$

where $\hat{c}_\sigma(\mathbf{k}) = \frac{1}{\sqrt{M}}\sum_{\mathbf{r}} e^{-i\mathbf{k}\cdot\mathbf{r}}\hat{c}_{\mathbf{r}\sigma}$. The set $\mathscr{F}$ consists of any of the wavevectors in the $1^{\text{st}}$ Brillouin zone. We denote the number of $\mathbf{k}$'s in $\mathscr{F}$ by $N_{\mathbf{k}}$. We normalize these states by computing $\hat{c}_N^{\mathbf{k}} = \sqrt{\frac{(M-N_{\mathbf{k}}-N)!}{(M-N_{\mathbf{k}})!N!}}$, where clearly $N + N_{\mathbf{k}} \leq M$. For large $M$, there are $\sim M^2 \times 2^{M-2}$ of such eigenstates [9]. Although this is a very large number, the total number of states in the Hilbert space is larger i.e. $4^M$. Thus the relative fraction of eta-pairing states vanishes as $M \to \infty$ [9]. The eigenenergy of $|\psi_N^{\{\mathbf{k}\}}\rangle$ is

$$E_{\psi_N^{\{\mathbf{k}\}}} = (U-2\mu)N + \sum_{\mathbf{k}\in\mathscr{F}}(\epsilon_{\mathbf{k}}-\mu), \tag{12}$$

where $\epsilon_{\mathbf{k}}$ are the energies of the kinetic term (i.e. in two dimensions $\epsilon_{\mathbf{k}} = 2t(\cos k_x + \cos k_y)$). Consider first the states in Eq. 11 with $N = 0$; all such states can be easily constructed, as they are non-interacting. For a given $N_{\mathbf{k}}$, such fully spin polarized states are highest weight spin states $S_z = S = \frac{N_{\mathbf{k}}}{2}$. They also have $J_0 = -J = \frac{1}{2}(N_{\mathbf{k}}-M)$, i.e. they are lowest weight pseudospin states.

The states in Eq. 11 are the $J_0 = \frac{1}{2}(2N+N_{\mathbf{k}}-M)$ states of the $J = \frac{1}{2}(M-N_{\mathbf{k}})$ pseudospin multiplet and highest weight spin states $S_z = S = \frac{N_{\mathbf{k}}}{2}$. Up to global spin $SU(2)$ rotations – obtained by repeated application of $\hat{S}_\pm$ – the states of Eq. 11 are the *only* states with $J + S = \frac{M}{2}$. If there were others, we could lower their $J_0$ by applying $\hat{J}_-$ $N$-times until we got to $J_0 = S - \frac{1}{2}M$.

From the definition of $\hat{J}_0$ below Eq. 10, this is a state with $2S$ spin-1/2 Fermions and total spin $S$ - therefore fully spin polarized. The only such states are non-interacting.

The reduced density matrix $\hat{\rho}_A^{\{k\}} = \text{Tr}_B\left(|\psi_N^{\{k\}}\rangle\langle\psi_N^{\{k\}}|\right)$ can be computed using the Schmidt decomposition of the Slater determinant part of Eq. 11,

$$\prod_{\mathbf{k}\in\mathscr{F}}\hat{c}_\downarrow^\dagger(\mathbf{k})|0\rangle = \prod_{m=1}^{N_\mathbf{k}}\left(\sqrt{\gamma_m}\hat{a}_m^\dagger + \sqrt{1-\gamma_m}\hat{b}_m^\dagger\right)|0\rangle. \tag{13}$$

Here

$$\hat{a}_m^\dagger = \frac{1}{\sqrt{\gamma_m}}\frac{1}{\sqrt{M}}\sum_{\mathbf{r}\in A}\left(\sum_{\mathbf{k}\in\mathscr{F}}e^{i\mathbf{k}\cdot\mathbf{r}}\phi_m^*(\mathbf{k})\right)\hat{c}_{\mathbf{r}\downarrow}^\dagger, \tag{14}$$

$$\hat{b}_m^\dagger = \frac{1}{\sqrt{1-\gamma_m}}\frac{1}{\sqrt{M}}\sum_{\mathbf{r}\in B}\left(\sum_{\mathbf{k}\in\mathscr{F}}e^{i\mathbf{k}\cdot\mathbf{r}}\phi_m^*(\mathbf{k})\right)\hat{c}_{\mathbf{r}\downarrow}^\dagger, \tag{15}$$

and $\gamma_m$ and $\phi_m(\mathbf{k})$ are respectively the eigenvalues and orthonormal eigenvectors of the Hermitian $N_\mathbf{k}\times N_\mathbf{k}$ matrix [17] $\Gamma_{\mathbf{kk'}} = \frac{1}{M}\sum_{\mathbf{r}\in A}e^{i(\mathbf{k}-\mathbf{k'})\cdot\mathbf{r}}$, with $\mathbf{k}$ and $\mathbf{k'}\in\mathscr{F}$. The Fermion operators in Eqs. 14-15 obey $\{\hat{a}_m^\dagger, \hat{a}_{m'}\} = \{\hat{b}_m^\dagger, \hat{b}_{m'}\} = \delta_{m,m'}$, and, because they live in different regions in real space, $\{\hat{a}_m^\dagger, \hat{b}_{m'}\} = 0$.

Using Eq. 13, we can write

$$\prod_{\mathbf{k}\in\mathscr{F}}\hat{c}_\downarrow^\dagger(\mathbf{k})|0\rangle = \sum_{\{m_A\}}\alpha_{\{m_A\}}|\{m_A\}\rangle\otimes|\{m_B\}\rangle, \tag{16}$$

where the sum is over all the different $2^{N_\mathbf{k}}$ ways to partition the $N_\mathbf{k}$ "orbitals" $m$ into those occupied by $a^\dagger$'s, denoted by the set $\{m_A\}$, and the complementary set $\{m_B\}$ occupied by $\hat{b}^\dagger$'s. If, for any given partition, there are $N_A$ "orbitals" in the set $\{m_A\}$, then there are $N_\mathbf{k} - N_A$ "orbitals" in the set $\{m_B\}$. Here,

$$\alpha_{\{m_A\}} = \left(\prod_{m\in\{m_A\}}\sqrt{\gamma_m}\right)\left(\prod_{m\in\{m_B\}}\sqrt{1-\gamma_m}\right). \tag{17}$$

The states in Eq. 16 are

$$|\{m_A\}\rangle = \prod_{m\in\{m_A\}}\hat{a}_m^\dagger|0\rangle, \qquad |\{m_B\}\rangle = \prod_{m\in\{m_B\}}\hat{b}_m^\dagger|0\rangle. \tag{18}$$

The reduced density matrix can again be calculated with the help of the contour integral representation of $|\psi_N^{\{k\}}\rangle$

$$\hat{\rho}_A^{\{k\}} = \sum_{\{m_A\}}\sum_{j=0}^{M_B-(N_\mathbf{k}-N_A)}\alpha_{\{m_A\}}^2\lambda_j^\mathbf{k}|N-j,\{m_A\}\rangle\langle N-j,\{m_A\}|,$$

$$\lambda_j^\mathbf{k} = \frac{\binom{M_B-(N_\mathbf{k}-N_A)}{j}\binom{M_A-N_A}{N-j}}{\binom{M-N_\mathbf{k}}{N}} \tag{19}$$

where $|N-j,\{m_A\}\rangle$ are orthonormal.

The von Neumann entanglement entropy is then

$$S_A^{\mathbf{k}} = -\sum_{\{m_A\}} \sum_{j=0}^{M_B-(N_{\mathbf{k}}-N_A)} \alpha_{\{m_A\}}^2 \lambda_j^{\mathbf{k}} \ln\left(\alpha_{\{m_A\}}^2 \lambda_j^{\mathbf{k}}\right). \tag{20}$$

Using the Vandermonde convolution, we can write the above as

$$S_A^{\mathbf{k}} = \tilde{S}_A^{\mathbf{k}} - \sum_{\{m_A\}} \sum_{j=0}^{M_B-(N_{\mathbf{k}}-N_A)} \alpha_{\{m_A\}}^2 \lambda_j^{\mathbf{k}} \ln \lambda_j^{\mathbf{k}}. \tag{21}$$

where $\tilde{S}_A^{\mathbf{k}} = -\sum_{\{m_A\}} \alpha_{\{m_A\}}^2 \ln \alpha_{\{m_A\}}^2$ is the von Neumann entropy of the free Fermi gas. As computed in Fig. 1 (for the two dimensional case) and as discussed by Lai and Yang [18], it can result in either the volume or the area-times-log "law", depending on which $\mathbf{k}$-points are occupied.

To analyze the second term in Eq. 21, we note that the sum over the partitions is sharply peaked around $N_A \approx \frac{N_{\mathbf{k}}}{M} M_A$, which is large in the limit of interest. Therefore, $\lambda_j^{\mathbf{k}}$ is peaked about a large value of $j$ and we can use Stirling's approximation. Again, replacing the discrete sum by an integral, we finally find

$$S_A^{\mathbf{k}} = \tilde{S}_A^{\mathbf{k}} + \frac{1}{2} + \frac{1}{2} \ln\left(2\pi\nu\left(1 - \frac{\nu}{1-\nu_{\mathbf{k}}}\right) M_A\right), \tag{22}$$

where $\nu_{\mathbf{k}} = N_{\mathbf{k}}/M$. We see that the leading order scaling comes from the free Fermion part (i.e. $\tilde{S}_A^{\mathbf{k}}$), and the correction scales with the logarithm of the number of sites in the region $A$. Logarithmically diverging subleading contribution was also argued for in Ref. [19], but with a different prefactor.

The Rényi entropy, $S_A^{\mathbf{k},(n)} = \frac{1}{1-n}\ln\left(\text{Tr}_A \hat{\rho}_A^n\right)$, can be computed for the states Eq. 11 using similar techniques. In the same limit as before, we find

$$S_A^{\mathbf{k},(n)} = \tilde{S}_A^{\mathbf{k},(n)} + \frac{1}{2}\frac{\ln n}{n-1} + \frac{1}{2}\ln\left(2\pi\nu\left(1 - \frac{\nu}{1-\nu_{\mathbf{k}}}\right)\frac{1-\nu_A^{(n)}}{1-\nu_{\mathbf{k}}} M_A\right). \tag{23}$$

where $\tilde{S}_A^{\mathbf{k},(n)}$ is the Rényi entropy of the free Fermi gas and

$$\nu_A^{(n)} = \frac{1}{M_A}\sum_m \frac{1}{\left(\frac{1}{\gamma_m}-1\right)^n + 1}. \tag{24}$$

Note that $\nu_A^{(1)} = \frac{1}{M_A}\text{Tr}\Gamma = \nu_{\mathbf{k}}$. The formulas Eq. 22 and Eq. 23 are therefore identical as $n \to 1$. Again, the leading scaling comes from the free Fermi part and the correction scales as $\sim \ln M_A$.

In the context of the ETH, it is interesting to ask whether the entanglement entropy density – be it von Neumann or Rényi – for the above mentioned exact eigenstates of the Hubbard model match the entropy density for the thermal density matrix $\hat{\rho}_{th} = e^{-\beta(\hat{H} - \mu_{th}^{(1)}\hat{J}_0 - \mu_{th}^{(2)}\hat{J})}$ with $\hat{H}$ being the Hubbard Hamiltonian. We included $\mu_{th}^{(1,2)}$ to separately control the average value of $N_{\mathbf{k}}$ and $N$. If the trace of $\hat{\rho}_{th}$ is to be performed over *all* the states in the Hilbert space of the Hubbard model, then they should not match, because $\text{Tr}\hat{\rho}_{th}$ should depend on the interaction $U$ while $\rho_A^{\{\mathbf{k}\}}$ is $U$-independent. However, if the trace is restricted to states of the type Eq. 11, and the distribution of the occupied $\mathbf{k}$ states results in the "volume" law (see Fig.1), then the first (leading) term in Eq. 23, indeed matches the "thermal" Rényi entropies computed in the grand canonical ensemble:

$$S_{th}^{(n)} = \frac{1}{1-n}\sum_{\mathbf{k}} \ln\left(f_{\mathbf{k}}^n + (1-f_{\mathbf{k}})^n\right), \tag{25}$$

$$f_{\mathbf{k}} = \frac{1}{e^{\beta(\epsilon_{\mathbf{k}}-\bar{\mu})} + 1}, \tag{26}$$

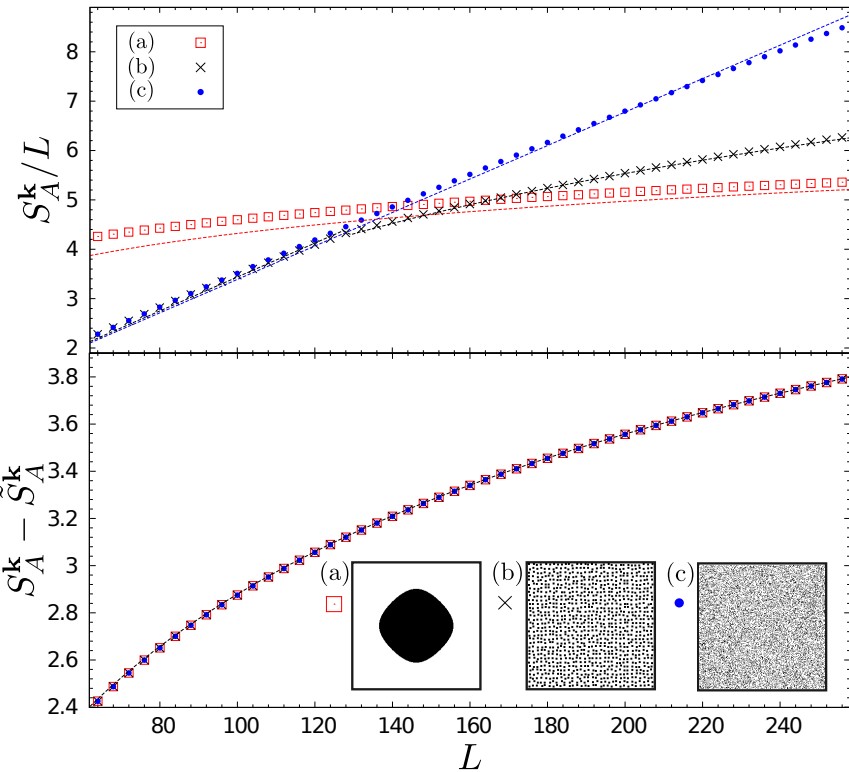

Figure 1: *Upper panel*: $S_A^{\mathbf{k}}$ computed from Eq. 21 for three different distributions in the Brillouin zone: a Fermi sea distribution (a), a regular pattern with a small random offset (b) and a fully random distribution (c). The distributions are shown as an inset of the lower panel, each black pixel being an occupied state. The system has $M = 256 \times 256$ sites with $N_{\mathbf{k}} = 16384$ particles and $N = 2048$ pairs. The region $A$ is a square of perimeter $L$. We actually show $S_A^{\mathbf{k}}/L$ and we have rescaled the values obtain for (a) by a factor of 10. The lines are the free Fermi gas entropies $\tilde{S}_A^{\mathbf{k}}$. Note that the distribution (b) has a crossover between area law (for small $L$) and $L \ln L$ (for large $L$). *Lower panel*: Difference between $S_A^{\mathbf{k}}$ and the von Neumann entropy of the free system $\tilde{S}_A^{\mathbf{k}}$ as a function of $L$ for the three distributions. The dashed line is the analytic difference given by Eq. 22.

| $n$ | $S_A^{\mathbf{k},(n)}/M_A$ | $S_{\text{th}}^{(n)}/M$ | $n$ | $S_A^{\mathbf{k},(n)}/M_A$ | $S_{\text{th}}^{(n)}/M$ |
|---|---|---|---|---|---|
| 1 | 0.560261 | 0.562334 | 6 | 0.345661 | 0.344944 |
| 2 | 0.468519 | 0.470002 | 7 | 0.336312 | 0.335553 |
| 3 | 0.412948 | 0.413339 | 8 | 0.329527 | 0.328758 |
| 4 | 0.379767 | 0.379486 | 9 | 0.324403 | 0.323636 |
| 5 | 0.359168 | 0.358576 | 10 | 0.320407 | 0.319645 |

Table 1: Rényi entropies per unit of volume. The second column is the Rényi entropy per unit of volume computed for a $16 \times 16$ patch using the same system than in Fig. 1 and the random distribution (c). The third column is the thermal Rényi entropy per unit of volume evaluated using the fitted parameters of $\beta$ and $\bar{\mu}$.

provided the values of $\beta$ and $\bar{\mu}$ are selected so that $\sum_{\mathbf{k}\in F}\epsilon_{\mathbf{k}} = \sum_{\mathbf{k}}\epsilon_{\mathbf{k}}f_{\mathbf{k}}$ and $N_{\mathbf{k}} = \sum_{\mathbf{k}}f_{\mathbf{k}}$. For the $\mathbf{k}$-distribution shown in Fig. 1, the comparison is shown in the Table 1. We note in passing that if the thermal Rényi entropy density is computed in the canonical ensemble, they do not match $S_A^{\mathbf{k},(n)}/M_A$ for $n > 1$.

## 4 Conclusion

In conclusion, we have obtained the exact closed-form expression for the entanglement spectrum of exact many-body excited eigenstates of the Hubbard model. Despite being exact excited eigenstates with finite energy density above the ground state, these states violate strong ETH. This is either because their entanglement entropy is sub-extensive or because it is interaction independent. Nevertheless, despite an exponentially large number of these states [9], the fraction of these state in the Hilbert space vanishes in the thermodynamic limit. As such, they may provide a useful starting point for studying the onset of thermalization due to perturbations which violate the total pseudospin conservation.

## Acknowledgements

We would like to acknowledge helpful conversations with Prof. Kun Yang. B. A. B. was supported by Department of Energy DE-SC0016239, Simons Investigator Award, ONR-N00014-14-1-0330, ARO MURI W911NF-12-1-0461, NSF-MRSEC DMR-1420541, Packard Foundation and Schmidt Fund for Innovative Research. N. R. was supported by Packard Foundation, MURI-130- 6082 and the Princeton Global Scholarship. O. V. was supported by NSF grant DMR 1506756.

## Appendix

In this Appendix, we provide detailed derivations of our results discussed in the main article. We also discusses extensions to the "generalized" Hubbard model and to a larger class of spin-flip eta-pairing states.

## A   Hubbard Hamiltonian and $SO(4)$ symmetry

In this section, we introduce the extended Hubbard model and its symmetries. We consider a Hamiltonian

$$\hat{H} = \hat{T} + \hat{V}_\beta, \tag{27}$$

where the kinetic energy $\hat{T}$ is

$$\hat{T} = \sum_{\mathbf{k},\sigma} (\epsilon_{\mathbf{k}} - \mu)(\hat{c}^\dagger_{\mathbf{k}\sigma} \hat{c}_{\mathbf{k}_\sigma}) \tag{28}$$

while the potential energy is a density-density "shifted" interaction:

$$\hat{V}_\beta = U \sum_{\mathbf{r}} \hat{c}^\dagger_{\mathbf{r}+\boldsymbol{\beta}\uparrow} \hat{c}_{\mathbf{r}+\boldsymbol{\beta}\uparrow} \hat{c}^\dagger_{\mathbf{r}\downarrow} \hat{c}_{\mathbf{r}\downarrow} \tag{29}$$

The case considered in the main text of the paper is $\boldsymbol{\beta} = 0$, but for now we keep a generic $\boldsymbol{\beta}$. For notation simplicity, bold symbols (such as $\mathbf{r}$ or $\boldsymbol{\beta}$) represent vectors in the $d$-dimensional space. We also define the shifted momentum

$$\hat{P} = \sum_{\mathbf{k}} \left(\mathbf{k} - \frac{1}{2}\mathbf{G}\right)(\hat{c}^\dagger_{\mathbf{k}\uparrow} \hat{c}_{\mathbf{k}\uparrow} + \hat{c}^\dagger_{\mathbf{k}\downarrow} \hat{c}_{\mathbf{k}\downarrow}) \tag{30}$$

where $\mathbf{G}$ is a given vector on the lattice, to be determined later.

## A.1 Spin symmetry

An extension of the usual $SU(2)$ spin symmetry exists in Eq. 27. We define the operator:

$$\hat{\zeta}_{\boldsymbol{\alpha}} = \sum_{\mathbf{r}} \hat{c}_{\mathbf{r}+\boldsymbol{\alpha}\uparrow} \hat{c}_{\mathbf{r}\downarrow}^{\dagger} = \sum_{\mathbf{k}} e^{i\mathbf{k}\cdot\boldsymbol{\alpha}} \hat{c}_{\mathbf{k}\uparrow} \hat{c}_{\mathbf{k}\downarrow}^{\dagger} \tag{31}$$

We have

$$[\hat{\zeta}_{\boldsymbol{\alpha}}^{\dagger}, \hat{\zeta}_{\boldsymbol{\alpha}}] = \sum_{\mathbf{r}} (\hat{c}_{\mathbf{r}\uparrow}^{\dagger} \hat{c}_{\mathbf{r}\uparrow} - \hat{c}_{\mathbf{r}\downarrow}^{\dagger} \hat{c}_{\mathbf{r}\downarrow}) \tag{32}$$

Using $[\hat{\zeta}_{\boldsymbol{\theta}}, [\hat{\zeta}_{\boldsymbol{\alpha}}^{\dagger}, \hat{\zeta}_{\boldsymbol{\alpha}}]] = 2\hat{\zeta}_{\boldsymbol{\theta}}$ we find that any of the $\hat{\zeta}_{\boldsymbol{\alpha}}$ operators and the $\hat{S}_z$ "spin" operator can form a $SU(2)$ algebra:

$$\hat{\zeta}_{\boldsymbol{\alpha}}^{\dagger} = \hat{S}_x + i\hat{S}_y, \quad \hat{S}_z = \frac{1}{2} \sum_{\mathbf{r}} (\hat{c}_{\mathbf{r}\uparrow}^{\dagger} \hat{c}_{\mathbf{r}\uparrow} - \hat{c}_{\mathbf{r}\downarrow}^{\dagger} \hat{c}_{\mathbf{r}\downarrow}) \tag{33}$$

$\hat{S}_z$ clearly commutes with $\hat{H}$ and for any $\boldsymbol{\alpha}$ we have $[\hat{\zeta}_{\boldsymbol{\alpha}}, \hat{T}] = [\hat{\zeta}_{\boldsymbol{\alpha}}, \hat{P}] = 0$ For $\boldsymbol{\alpha} = \boldsymbol{\beta}$, the $\hat{\zeta}_{\boldsymbol{\beta}}$ operators also commutes with the density density part of the Hamiltonian:

$$[\hat{\zeta}_{\boldsymbol{\beta}}, \hat{V}_{\boldsymbol{\beta}}] = 0 \tag{34}$$

$\hat{\zeta}_{\boldsymbol{\beta}}, \hat{\zeta}_{\boldsymbol{\beta}}^{\dagger}, \hat{S}_z$ form an $SU(2)$ spin algebra.

## A.2 $\eta$ symmetry

We now define an $\hat{\eta}$ operator:

$$\hat{\eta}_{\boldsymbol{\alpha}} = \sum_{\mathbf{r}} e^{-i\mathbf{G}\cdot\mathbf{r}} \hat{c}_{\mathbf{r}+\boldsymbol{\alpha}\uparrow} \hat{c}_{\mathbf{r}\downarrow} = \sum_{\mathbf{k}} e^{i\mathbf{k}\cdot\boldsymbol{\alpha}} \hat{c}_{\mathbf{k}\uparrow} \hat{c}_{\mathbf{G}-\mathbf{k}\downarrow} \tag{35}$$

with an algebra:

$$[\hat{\eta}_{\boldsymbol{\alpha}}^{\dagger}, \hat{\eta}_{\boldsymbol{\alpha}}] = \sum_{\mathbf{r}} (\hat{c}_{\mathbf{r}\uparrow}^{\dagger} \hat{c}_{\mathbf{r}\uparrow} + \hat{c}_{\mathbf{r}\downarrow}^{\dagger} \hat{c}_{\mathbf{r}\downarrow}) - M \tag{36}$$

where M is the total number of sites in the problem. The general commutation relation $[\hat{\eta}_{\boldsymbol{\gamma}}, [\hat{\eta}_{\boldsymbol{\theta}}^{\dagger}, \hat{\eta}_{\boldsymbol{\alpha}}]] = 2\hat{\eta}_{\boldsymbol{\alpha}+\boldsymbol{\gamma}-\boldsymbol{\theta}}$ means that any of the $\hat{\eta}_{\boldsymbol{\alpha}}$ operators and the number of particle operator form an $SU(2)$ algebra:

$$\hat{\eta}_{\boldsymbol{\alpha}}^{\dagger} = \hat{J}_x + i\hat{J}_y, \quad \hat{J}_z = \frac{1}{2} \sum_{\mathbf{r}} (\hat{c}_{\mathbf{r}\uparrow}^{\dagger} \hat{c}_{\mathbf{r}\uparrow} + \hat{c}_{\mathbf{r}\downarrow}^{\dagger} \hat{c}_{\mathbf{r}\downarrow}) - \frac{1}{2}M \tag{37}$$

with the usual $[\hat{J}_x, \hat{J}_y] = i\hat{J}_z$, relations of the $SU(2)$ algebra. This algebra is true for any $\boldsymbol{\alpha}$. The $\hat{\eta}_{\boldsymbol{\alpha}}, \hat{\zeta}_{\boldsymbol{\alpha}}$ operators commute, forming an $SU(2) \times SU(2)$ algebra

$$[\hat{\eta}_{\boldsymbol{\alpha}}, \hat{\zeta}_{\boldsymbol{\alpha}}] = [\hat{\eta}_{\boldsymbol{\alpha}}^{\dagger}, \hat{\zeta}_{\boldsymbol{\alpha}}] = [\hat{\eta}_{\boldsymbol{\alpha}}^{\dagger}, \hat{J}_z] = [\hat{\zeta}_{\boldsymbol{\alpha}}^{\dagger}, \hat{S}_z] = 0 \tag{38}$$

For any $\mathbf{G}$, we have $[\hat{\eta}_{\boldsymbol{\alpha}}, \hat{P}] = 0$. We now check the general conditions when $\hat{\eta}_{\boldsymbol{\alpha}}$ has interesting commutation relations with the Hamiltonian. For the kinetic term of the Hamiltonian we find:

$$[\hat{\eta}_{\boldsymbol{\alpha}}, \hat{T}] = \sum_{\mathbf{k}} e^{i\mathbf{k}\cdot\boldsymbol{\alpha}} \hat{c}_{\mathbf{k}\uparrow} \hat{c}_{\pi-\mathbf{k}\downarrow} (\epsilon_{\mathbf{k}} + \epsilon_{\mathbf{G}-\mathbf{k}} + 2\mu) \tag{39}$$

For any $(\epsilon_{\mathbf{k}} + \epsilon_{\mathbf{G}-\mathbf{k}})$ independent of $\mathbf{k}$ the right hand side is just $\hat{\eta}_{\boldsymbol{\alpha}}$. For the nearest neighbor (or any "odd" neighbor) hopping where $\epsilon_{\mathbf{k}} = 2t \sum_{i=x,y,\dots} \cos(k_i)$ and hence

$$\mathbf{G} = \pi \tag{40}$$

where $\boldsymbol{\pi}$ is a $d$-dimensional vector with all components equal to $\pi$. In that case, we have $[\hat{\eta}_{\boldsymbol{\alpha}}, \hat{T}] = 2\mu\hat{\eta}_{\boldsymbol{\alpha}}$. However, only $\hat{\eta}_{\boldsymbol{\beta}}$ (i.e. for $\boldsymbol{\alpha} = \boldsymbol{\beta}$) commutes with the potential part $\hat{V}_{\boldsymbol{\beta}}$ from Eq. 29:

$$[\hat{\eta}_{\boldsymbol{\beta}}, \hat{V}_{\boldsymbol{\beta}}] = U\hat{\eta}_{\boldsymbol{\beta}} \tag{41}$$

Hence:

$$[\hat{\eta}_{\boldsymbol{\beta}}, \hat{H}] = (2\mu + U)\hat{\eta}_{\boldsymbol{\beta}} \tag{42}$$

We now have proved that $\hat{\eta}_{\boldsymbol{\beta}}, \hat{\eta}_{\boldsymbol{\beta}}^{\dagger}, \hat{J}_z$ and $\hat{\zeta}_{\boldsymbol{\beta}}, \hat{\zeta}_{\boldsymbol{\beta}}^{\dagger}, \hat{S}_z$ form and $SU(2) \times SU(2)$ symmetry generators that inter-commute. $\hat{\eta}_{\boldsymbol{\beta}}$ also has a nice commutation with the Hamiltonian $\hat{H}$. In fact, we can shift $\mu = -U/2$ and have $\hat{\eta}_{\boldsymbol{\beta}}$ commute with the Hamiltonian $\hat{T} + \hat{V}_{\boldsymbol{\beta}}$.

# B  Explicit eigenfunctions of $\hat{H}$

The spectrum of $\hat{H}$ can be placed in eigenvalues of $\hat{J}^2, \hat{J}_z, \hat{S}^2, \hat{S}_z, \hat{H}, \hat{P}$. We will now write down a large number of exact eigenstates. Out of $\hat{J}_z, \hat{S}_z$ we can make two (linearly dependent operators) quantum numbers, $\hat{N}_{\uparrow} = \sum_{\mathbf{r}} \hat{c}_{\mathbf{r}\uparrow}^{\dagger} \hat{c}_{\mathbf{r}\uparrow}$ and $\hat{N}_{\downarrow} = \sum_{\mathbf{r}} \hat{c}_{\mathbf{r}\downarrow}^{\dagger} \hat{c}_{\mathbf{r}\downarrow}$:

$$\hat{J}_z = \frac{\hat{N}_{\uparrow} + \hat{N}_{\downarrow} - M}{2}, \quad \hat{S}_z = \frac{\hat{N}_{\uparrow} - \hat{N}_{\downarrow}}{2} \tag{43}$$

The numbers of $\uparrow$ and $\downarrow$ spins are independently conserved and will be used to interchangeably denote states.

## B.1  Set of eigenstates

First consider the eigenstates of the Hamiltonian for which the number $N_{\uparrow}$ of $\uparrow$ particles is zero. For these states, the interaction do not appear and we have $N_{\downarrow} = N_{\mathbf{k}}$ noninteracting fermions, each with their momenta:

$$\left|\Psi_{0,0}^{\{\mathbf{k}\}}\right\rangle = \left|k_1, \ldots, k_{N_{\mathbf{k}}}\right\rangle = \prod_{\mathbf{k}\in\mathcal{F}} \hat{c}_{\downarrow}^{\dagger}(\mathbf{k}) |0\rangle \tag{44}$$

where $\mathcal{F}$ consists of any set of $N_{\mathbf{k}}$ wavevectors in the 1st Brillouin zone. The energy and momentum of these states is:

$$E_{\left|\Psi_{0,0}^{\{\mathbf{k}\}}\right\rangle} = \sum_{\mathbf{k}\in\mathcal{F}} \epsilon_{\mathbf{k}} - \mu N_{\mathbf{k}} \tag{45}$$

$$P_{\left|\Psi_{0,0}^{\{\mathbf{k}\}}\right\rangle} = \sum_{\mathbf{k}\in\mathcal{F}} \mathbf{k} - \frac{1}{2} N_{\mathbf{k}} \boldsymbol{\pi} \mod 2\pi \tag{46}$$

We can see that these states have:

$$\hat{\eta}_{\boldsymbol{\beta}} \left|\Psi_{0,0}^{\{\mathbf{k}\}}\right\rangle = 0, \quad \hat{\zeta}_{\boldsymbol{\beta}} \left|\Psi_{0,0}^{\{\mathbf{k}\}}\right\rangle = 0,$$

$$\hat{J}_z \left|\Psi_{0,0}^{\{\mathbf{k}\}}\right\rangle = \frac{N_{\mathbf{k}}-M}{2} \left|\Psi_{0,0}^{\{\mathbf{k}\}}\right\rangle, \quad \hat{S}_z \left|\Psi_{0,0}^{\{\mathbf{k}\}}\right\rangle = -\frac{N_{\mathbf{k}}}{2} \left|\Psi_{0,0}^{\{\mathbf{k}\}}\right\rangle \tag{47}$$

and hence these are the lowest weight states of a multiplet:

$$\left|\Psi_{N_1,N_2}^{\{\mathbf{k}\}}\right\rangle = (\hat{\eta}_{\boldsymbol{\beta}}^{\dagger})^{N_1} (\hat{\zeta}_{\boldsymbol{\beta}}^{\dagger})^{N_2} \left|\Psi_{0,0}^{\{\mathbf{k}\}}\right\rangle \tag{48}$$

with $N_1 = 0, \ldots, M - N_{\mathbf{k}}$, $N_2 = 0, \ldots, N_{\mathbf{k}}$. These states have the quantum numbers under $\hat{H}, \hat{P}, \hat{J}_z, \hat{S}_z$ respectively:

$$E_{\left|\Psi_{0,0}^{\{\mathbf{k}\}}\right\rangle} - (\mu - \frac{1}{2}U)N_1, \quad P_{\left|\Psi_{0,0}^{\{\mathbf{k}\}}\right\rangle}, \quad \frac{N_{\mathbf{k}}-M}{2} + N_1, \quad -\frac{N_{\mathbf{k}}}{2} + N_2 \tag{49}$$

This is a large number of states: the different $\mathscr{F}$ configurations are $\begin{pmatrix} M \\ N_\downarrow \end{pmatrix}$ for a total of

$$\sum_{N_\downarrow=0}^{M} \begin{pmatrix} M \\ N_\downarrow \end{pmatrix} (M - N_\downarrow + 1)(N_\downarrow + 1). \tag{50}$$

The state in Eq. 11 of the main text, is the $N_1 = N, N_2 = 0, N_\mathbf{k} = N_\downarrow$ representative of Eq. 48 and was first introduced by C. N. Yang in Ref. [7].

In Ref. [7], C.N. Yang built the one state $|N_1, 0, 0\rangle$ of the set Eq. 48, the so-called eta-pairing states. He then proceeded to building another set of states, $\hat{\eta}_{\boldsymbol\beta}^N \hat{\eta}_{\boldsymbol\alpha}^\dagger |0\rangle$ which he then proved were also eigenstates of the Hamiltonian. This state is, however, linearly dependent on $\left| \Psi_{N_1,N_2}^{\{\mathbf{k}\}} \right\rangle$ and hence not a new state. CN Yang's state $|\boldsymbol\alpha\rangle = \hat{\eta}_{\boldsymbol\alpha}^\dagger |0\rangle$, $\forall \, \boldsymbol\alpha \neq \boldsymbol\beta$ has the $J_z, S_z$ quantum numbers $\hat{J}_z |\boldsymbol\alpha\rangle = \frac{2-M}{2} |\boldsymbol\alpha\rangle$, $\hat{S}_z |\boldsymbol\alpha\rangle = 0$. It belongs to a multiplet $\left| N_3, N_4, \boldsymbol\alpha \right\rangle = (\hat{\eta}_{\boldsymbol\beta}^\dagger)^{N_3} (\hat{\zeta}_{\boldsymbol\beta}^\dagger)^{N_4} |\boldsymbol\alpha\rangle$ where and $N_3 = 0, \dots, M-2$ and $N_4 = -1, 0, 1$ (the notation is $(\hat{\zeta}_{\boldsymbol\beta}^\dagger)^{-1} = \hat{\zeta}_{\boldsymbol\beta}$). By direct calculation we find that the lowest weight states $|0, -1, \boldsymbol\alpha\rangle$ *are* (an energy $-2\mu$) linear combination of the states $\left| \Psi_{N_1,N_2}^{\{\mathbf{k}\}} \right\rangle = |0, 0; k, \pi - k\rangle$ of Eq. 48:

$$\hat{\zeta}_{\boldsymbol\beta} |\boldsymbol\alpha\rangle = \sum_{\mathbf{k}} e^{i(\boldsymbol\beta - \boldsymbol\alpha)\cdot\mathbf{k}} \hat{c}_\downarrow^\dagger(\mathbf{k}) \hat{c}_\downarrow^\dagger(\boldsymbol\pi - \mathbf{k}) |0\rangle \tag{51}$$

This in fact had to be so because we will now prove that the states Eq. 48 are the only states (and their $\eta$ and $\hat{\zeta}$) in the $J + S = M/2$ sector. Since $\left| N_3, N_4, \boldsymbol\alpha \right\rangle$ also have $J + S = M/2$, they must hence be a linear combination of the states in Eq. 48 .

## B.2 Completeness

The states in Eq. 48 have $J, S$ quantum numbers that satisfy the relation $J + S = M/2$: given a configuration of momenta $\mathscr{F}$, they have the same $J, S$ quantum numbers Eq. 47 as their lowest weight counterparts Eq. 44, which immediately satisfy the aforementioned identity. We now prove that *all the states* with $J + S = M/2$ are part of multiplets where the lowest (and highest) weight states are *noninteracting*. In other words, the set of states in Eq. 48 saturate the Hilbert space of quantum numbers $J + S = M/2$ (irrespective of $J_z, S_z, P$)

Pick any states $|J, S, J_z, S_z, P\rangle$ in the Hilbert space of the Hubbard model, with $J + S = M/2$. We can always apply the $\hat{\eta}_{\boldsymbol\beta}, \hat{\zeta}_{\boldsymbol\beta}$ lowering operators the appropriate amount of times to bring this state to the lowest weight of both $SU(2) \otimes SU(2)$: $|J, S, J_z = -J, S_z = -S, P\rangle$. For this last state, we have $J_z + S_z = -(J + S) = -M/2$. Eq. 43 relates the quantum numbers to the number of $\uparrow, \downarrow$ particles existent in the system. It is then trivial to see that the states $|J, S, J_z = -J, S_z = -S, P\rangle$ has $N_\uparrow = 0$ (without any restriction on $N_\downarrow$). Since only $N_\downarrow$ particles are present, there is no Hubbard $U$ interaction, and *all* the lowest weight states $|J, S, J_z = -J, S_z = -S, P\rangle$ $(J + S = M/2)$ can be labeled by the momenta of the $\downarrow$ particle, as in Eq. 44. No other states can exist in these quantum number sectors.

## B.3 Norm of the $\left| \Psi_{N_1,N_2}^{\{\mathbf{k}\}} \right\rangle$ states

To fully define the states, we compute their norm. We first present a method which will be used extensively in the calculations in this section. First, we note that we can write a Kronecker $\delta$-function using contour integration as

$$\delta_{m,n} = \oint_C \frac{dz}{2\pi i} \frac{1}{z^{n+1}} z^m \tag{52}$$

where the contour $C$ encircles the origin in the complex $z$-plane counterclockwise. Using this and the commutation relations $[\hat{c}^\dagger_{\mathbf{r}'\downarrow}\hat{c}^\dagger_{\mathbf{r}'+\boldsymbol{\beta}\uparrow},\hat{c}_{\mathbf{r}\downarrow}\hat{c}^\dagger_{\mathbf{r}+\boldsymbol{\beta}\uparrow}]=0$ we re-write the states:

$$
\begin{aligned}
\left|\Psi^{\{\mathbf{k}\}}_{N_1,N_2}\right\rangle &= N_1!N_2!\oint\oint\frac{dz_1}{2\pi i}\frac{dz_2}{2\pi i}\frac{1}{z_1^{N_1+1}}\frac{1}{z_2^{N_2+1}}\sum_{n_1=0}^{\infty}\frac{1}{n_1!}(z_1\sum_{r'}e^{i\pi\cdot\mathbf{r}'}\hat{c}^\dagger_{\mathbf{r}'\downarrow}\hat{c}^\dagger_{\mathbf{r}'+\boldsymbol{\beta}\uparrow})^{n_1}\\
&\quad\times\sum_{n_2=0}^{\infty}\frac{1}{n_2!}(z_2\sum_{\mathbf{r}}\hat{c}_{\mathbf{r}\downarrow}\hat{c}^\dagger_{\mathbf{r}+\boldsymbol{\beta}\uparrow})^{n_2}\left|\Psi^{\{\mathbf{k}\}}_{0,0}\right\rangle\\
&= N_1!N_2!\oint\oint\frac{dz_1}{2\pi i}\frac{dz_2}{2\pi i}\frac{1}{z_1^{N_1+1}}\frac{1}{z_2^{N_2+1}}e^{z_1\sum_{r'}e^{i\pi\cdot\mathbf{r}'}\hat{c}^\dagger_{\mathbf{r}'\downarrow}\hat{c}^\dagger_{\mathbf{r}'+\boldsymbol{\beta}\uparrow}}e^{z_2\sum_{\mathbf{r}}\hat{c}_{\mathbf{r}\downarrow}\hat{c}^\dagger_{\mathbf{r}+\boldsymbol{\beta}\uparrow}}\left|\Psi^{\{\mathbf{k}\}}_{0,0}\right\rangle\\
&= N_1!N_2!\oint\oint\frac{dz_1}{2\pi i}\frac{dz_2}{2\pi i}\frac{1}{z_1^{N_1+1}}\frac{1}{z_2^{N_2+1}}\prod_{r'}(1+z_1 e^{i\pi\cdot\mathbf{r}'}\hat{c}^\dagger_{\mathbf{r}'\downarrow}\hat{c}^\dagger_{\mathbf{r}'+\boldsymbol{\beta}\uparrow})\\
&\quad\times\prod_{\mathbf{r}}(1+z_2\hat{c}_{\mathbf{r}\downarrow}\hat{c}^\dagger_{\mathbf{r}+\boldsymbol{\beta}\uparrow})\left|\Psi^{\{\mathbf{k}\}}_{0,0}\right\rangle\\
&= N_1!N_2!\oint\oint\frac{dz_1}{2\pi i}\frac{dz_2}{2\pi i}\frac{1}{z_1^{N_1+1}}\frac{1}{z_2^{N_2+1}}\prod_{\mathbf{r}}(1+z_1 e^{i\pi\cdot\mathbf{r}}\hat{c}^\dagger_{\mathbf{r}\downarrow}\hat{c}^\dagger_{\mathbf{r}+\boldsymbol{\beta}\uparrow}+z_2\hat{c}_{\mathbf{r}\downarrow}\hat{c}^\dagger_{\mathbf{r}+\boldsymbol{\beta}\uparrow})\left|\Psi^{\{\mathbf{k}\}}_{0,0}\right\rangle\quad(53)
\end{aligned}
$$

The product over $\mathbf{r}$ is taken over all lattice sites. We are now in a position to calculate the norm. Using the fact that $\left|\Psi^{\{\mathbf{k}\}}_{0,0}\right\rangle$ contains only $N_{\mathbf{k}}$ $b$- particles, and with the help of the identity

$$
e^{\alpha\hat{c}^\dagger_{\mathbf{r}\downarrow}\hat{c}_{\mathbf{r}\downarrow}}=1+(e^\alpha-1)\hat{c}^\dagger_{\mathbf{r}\downarrow}\hat{c}_{\mathbf{r}\downarrow},\tag{54}
$$

we find

$$
\begin{aligned}
\left\langle\Psi^{\{\mathbf{k}'\}}_{N_1',N_2'}\big|\Psi^{\{\mathbf{k}\}}_{N_1,N_2}\right\rangle &= N_1'!N_2'!N_1!N_2!\oint\oint\oint\oint\frac{dz_1}{2\pi i}\frac{dz_2}{2\pi i}\frac{dz_3}{2\pi i}\frac{dz_4}{2\pi i}\frac{1}{z_1^{N_1+1}}\frac{1}{z_2^{N_2+1}}\frac{1}{z_3^{N_1'+1}}\frac{1}{z_2^{N_2'+1}}\\
&\quad\times\left\langle\Psi^{\{\mathbf{k}'\}}_{N_1',N_2'}\Big|\prod_{\mathbf{r}}(1+z_1 z_3\hat{c}_{\mathbf{r}\downarrow}\hat{c}^\dagger_{\mathbf{r}\downarrow}+z_2 z_4\hat{c}^\dagger_{\mathbf{r}\downarrow}\hat{c}_{\mathbf{r}\downarrow})\Big|\Psi^{\{\mathbf{k}\}}_{0,0}\right\rangle\quad(55)
\end{aligned}
$$

The integrand can be massaged

$$
\begin{aligned}
\left\langle\Psi^{\{\mathbf{k}'\}}_{N_1',N_2'}\Big|\prod_{\mathbf{r}}(1+z_1 z_3\hat{c}_{\mathbf{r}\downarrow}\hat{c}^\dagger_{\mathbf{r}\downarrow}+z_2 z_4\hat{c}^\dagger_{\mathbf{r}\downarrow}\hat{c}_{\mathbf{r}\downarrow})\Big|\Psi^{\{\mathbf{k}\}}_{0,0}\right\rangle &= (1+z_1 z_3)^M\left\langle\Psi^{\{\mathbf{k}'\}}_{0,0}\Big|\prod_{\mathbf{r}}e^{\log(1+\frac{z_2 z_4-z_1 z_3}{1+z_1 z_3})\hat{c}^\dagger_{\mathbf{r}\downarrow}\hat{c}_{\mathbf{r}\downarrow}}\Big|\Psi^{\{\mathbf{k}\}}_{0,0}\right\rangle\\
&= (1+z_1 z_3)^M\left\langle\Psi^{\{\mathbf{k}'\}}_{0,0}\Big|e^{\log(\frac{1+z_2 z_4}{1+z_1 z_3})\sum_{\mathbf{r}}\hat{c}^\dagger_{\mathbf{r}\downarrow}\hat{c}_{\mathbf{r}\downarrow}}\Big|\Psi^{\{\mathbf{k}\}}_{0,0}\right\rangle\\
&= (1+z_1 z_3)^M e^{\log(\frac{1+z_2 z_4}{1+z_1 z_3})N_{\mathbf{k}}}\left\langle\Psi^{\{\mathbf{k}'\}}_{0,0}\big|\Psi^{\{\mathbf{k}\}}_{0,0}\right\rangle\\
&= (1+z_1 z_3)^{M-N_{\mathbf{k}}}(1+z_2 z_4)^{N_{\mathbf{k}}}\delta_{\mathscr{F}',\mathscr{F}}\quad(56)
\end{aligned}
$$

and provides the first Kronecker delta function of the momenta configurations $\mathscr{F}$ and $\mathscr{F}'$. Simple integration then provides for:

$$
\left\langle\Psi^{\{\mathbf{k}'\}}_{N_1',N_2'}\big|\Psi^{\{\mathbf{k}\}}_{N_1,N_2}\right\rangle = \frac{N_{\mathbf{k}}!N_2!(M-N_{\mathbf{k}})!N_1!}{(N_{\mathbf{k}}-N_2)!(M-N_{\mathbf{k}}-N_1)!}\delta_{N_1',N_1}\delta_{N_2',N_2}\delta_{\mathscr{F}',\mathscr{F}}\tag{57}
$$

## C  Entanglement spectrum of $\left|\Psi^{\{\mathbf{k}\}}_{N_1,N_2}\right\rangle$ states

The entanglement spectrum of the states $\left|\Psi^{\{\mathbf{k}\}}_{N_1,N_2}\right\rangle$ can be analytically computed for $\boldsymbol{\beta}=0$. We sketch this calculation in the current section. The strategy to diagonalized the reduced density

matrix will be to first perform the Schmidt decomposition of the non-interacting lowest weight states $\left|\Psi_{0,0}^{\{\mathbf{k}\}}\right\rangle$. In this basis, we then form the states $\left|\Psi_{N_1,N_2}^{\{\mathbf{k}\}}\right\rangle$ and apply techniques similar to those of Eq. 53 and Eq. 57 to diagonalize the density matrix. In all our calculations, we will assume that a partition of the space of $M$ sites has been performed in an $A$ and $B$ parts.

## C.1 Schmidt decomposition of $\left|\Psi_{0,0}^{\{\mathbf{k}\}}\right\rangle$

The strategy for performing a Schmidt decomposition of a non-interacting state $\left|\Psi_{0,0}^{\{\mathbf{k}\}}\right\rangle = \prod_{\mathbf{k}\in\mathscr{F}} \hat{c}_{\mathbf{k}\downarrow}^\dagger |0\rangle$ into $A$ (left) and $B$ (right) regions is well known and has been first presented by Peschel in Ref. [17]. We build and diagonalize the one-body density matrix of the $A$ side:

$$\Gamma_{\mathbf{k},\mathbf{k}'} = \frac{1}{M}\sum_{\mathbf{r}\in A} e^{-i(\mathbf{k}-\mathbf{k}')\mathbf{r}} \qquad \text{and} \qquad \sum_{\mathbf{k}'\in\mathscr{F}} \Gamma_{\mathbf{k},\mathbf{k}'}\phi_m(\mathbf{k}') = \gamma_m\phi_m(\mathbf{k}), \quad \mathbf{k}\in\mathscr{F}$$

where we normalize our complete basis: $\sum_{\mathbf{k}\in\mathscr{F}} \phi_m(\mathbf{k})^\star \phi_{m'}(\mathbf{k}) = \delta_{mm'}$. Any eigenvalues which are $0,1$ and their respective eigenstates are discarded. Using this complete basis we want to build eigenstates with support fully in either region $A$ or $B$. If for any $\gamma_m \neq 0,1$, we rescale

$$\phi_m(\mathbf{r}) = \frac{1}{\sqrt{M}\sqrt{\gamma_m}}\sum_{\mathbf{k}\in\mathscr{F}} e^{i\mathbf{k}\cdot\mathbf{r}}\phi_m(\mathbf{k}), \quad \mathbf{r}\in A \tag{58}$$

and

$$\phi_m(\mathbf{r}) = \frac{1}{\sqrt{M}\sqrt{1-\gamma_m}}\sum_{\mathbf{k}\in\mathscr{F}} e^{i\mathbf{k}\cdot\mathbf{r}}\phi_m(\mathbf{k}), \quad \mathbf{r}\in B \tag{59}$$

we have found normalized operators in the $A$ and $B$ side of the system:

$$\begin{aligned}
\sum_{\mathbf{r}\in A}\phi_m^\star(\mathbf{r})\phi_{m'}(\mathbf{r}) &= \frac{1}{M\gamma_m}\sum_{\mathbf{k},\mathbf{k}'\in\mathscr{F}}\left(\phi_m(\mathbf{k})^\star\phi_{m'}(\mathbf{k})\sum_{\mathbf{r}\in A}e^{-i(\mathbf{k}-\mathbf{k}')\cdot\mathbf{r}}\right) \\
&= \frac{1}{\gamma_m}\sum_{\mathbf{k},\mathbf{k}'\in\mathscr{F}}\phi_m(\mathbf{k})^\star\phi_{m'}(\mathbf{k})\Gamma_{\mathbf{k}\mathbf{k}'} \\
&= \sum_{\mathbf{k}\in\mathscr{F}}\phi_m(\mathbf{k})\phi_{m'}(\mathbf{k}) = \delta_{m,m'}
\end{aligned} \tag{60}$$

And similarly for the $B$ region.

We are now ready to Schmidt decompose the state. As $\phi_m(\mathbf{k})$ is a unitary transformation (keep all the $\gamma_m$'s, even if $0,1$), we perform the canonical transformation:

$$\hat{c}_m^\dagger = \sum_{\mathbf{k}\in\mathscr{F}}\phi_m(\mathbf{k})\hat{c}_{\mathbf{k}}^\dagger \tag{61}$$

which keep the state $\left|\Psi_{0,0}^{\{\mathbf{k}\}}\right\rangle$ invariant:

$$\left|\Psi_{0,0}^{\{\mathbf{k}\}}\right\rangle = \prod_{m=1}\hat{c}_m^\dagger |0\rangle \tag{62}$$

We now separate $\hat{c}_m^\dagger$ into orthogonal left and right second quantized operators:

$$
\begin{aligned}
\hat{c}_m^\dagger &= \sum_{\mathbf{k}\in\mathscr{F}} \phi_m(\mathbf{k}) c^\dagger(\mathbf{k}) \\
&= \frac{1}{M} \sum_{\mathbf{r}\in A+B} \sum_{\mathbf{k}\in\mathscr{F}} e^{ik\cdot j} \phi_m(\mathbf{k}) \hat{c}_\mathbf{r}^\dagger \\
&= \sqrt{\gamma_m} \sum_{\mathbf{r}\in A} \phi_m(\mathbf{r}) \hat{c}_\mathbf{r}^\dagger + \sqrt{1-\gamma_m} \sum_{\mathbf{r}\in B} \phi_m(\mathbf{r}) \hat{c}_\mathbf{r}^\dagger \\
&= \sqrt{\gamma_m} a_m^\dagger + \sqrt{1-\gamma_m} b_m^\dagger
\end{aligned}
\tag{63}
$$

Where $a_m = \sum_{\mathbf{r}\in A} \phi_m(\mathbf{r}) \hat{c}_\mathbf{r}$, $b_m = \sum_{\mathbf{r}\in B} \phi_m(\mathbf{r}) \hat{c}_\mathbf{r}$ are canonical fermionic operators with support exclusively on the left and right hand side respectively. Hence:

$$
\left| \Psi_{0,0}^{\{\mathbf{k}\}} \right\rangle = \prod_{m=1} (\sqrt{\gamma_m} a_m^\dagger + \sqrt{1-\gamma_m} b_m^\dagger) |0\rangle
\tag{64}
$$

No ↑ fermions are present in the state. The many-body Schmidt decomposition of the state can be decomposed in sectors that contain $N_A$ particles in the $A$ side and $N_\mathbf{k} - N_A$ particles on the $B$ side. Each $N_A$ sector on the $A$ side can be obtained by filling a set of $\{m_A\}$ single-particle eigenstates $m$. Written like this, the state is easily decomposed in

$$
\left| \Psi_{0,0}^{\{\mathbf{k}\}} \right\rangle = \sum_{\{m_A\}} \alpha_{\{m_A\}} |\{m_A\}\rangle \otimes |\{m_B\}\rangle,
\tag{65}
$$

where the sum is over all the different $2^{N_\mathbf{k}}$ ways to partition the $N_\mathbf{k}$ ↓-"orbitals" $m$ into those occupied by $a^\dagger$'s, denoted by the set $\{m_A\}$ - for the $N_A$ particle state $|\{m_A\}\rangle = \prod_{m\in\{m_A\}} a_m^\dagger |0\rangle$, and the complementary set $\{m_B\}$ occupied by $\hat{b}^\dagger$'s for the $N_\mathbf{k} - N_A$-particle state $|\{m_B\}\rangle = \prod_{m\in\{m_B\}} \hat{b}_m^\dagger |0\rangle$. Here

$$
\alpha_{\{m_A\}} = \left( \prod_{m\in\{m_A\}} \sqrt{\gamma_m} \right) \left( \prod_{m\in\{m_B\}} \sqrt{1-\gamma_m} \right)
\tag{66}
$$

The entanglement entropy is then

$$
\sum_{\{m_A\}} \alpha_{\{m_A\}}^2 \log \alpha_{\{m_A\}}^2 = \sum_m \gamma_m \log \gamma_m + (1-\gamma_m) \log(1-\gamma_m)
\tag{67}
$$

For the below, it is important to remember that $N_A$ is a good quantum number of the decomposition.

## C.2   Entanglement spectrum of all the $\left| \Psi_{N_1,N_2}^{\{\mathbf{k}\}} \right\rangle$ states for $\beta = 0$

Having obtained an $A/B$ decomposition of the states $\left| \Psi_{0,0}^{\{\mathbf{k}\}} \right\rangle$, we now obtain the decomposition for the full states $\left| \Psi_{N_1,N_2}^{\{\mathbf{k}\}} \right\rangle$. We start by building a new orthonormal basis for the $A$ and $B$ sides away from the lowest weight limit. We build the $\hat{\eta}$ and $\hat{\zeta}$ operators on the $A$ and $B$ sides respectively:

$$
\hat{\eta}_{A/B}^\dagger = \sum_{\mathbf{r}\in A/B} e^{i\pi\cdot\mathbf{r}} \hat{c}_{\mathbf{r}\downarrow}^\dagger \hat{c}_{\mathbf{r}\uparrow}^\dagger, \quad \hat{\zeta}_{A/B}^\dagger = \sum_{\mathbf{r}\in A/B} \hat{c}_{\mathbf{r}\downarrow}^\dagger \hat{c}_{\mathbf{r}\uparrow}^\dagger,
\tag{68}
$$

Using the Schmidt decomposition of left and right parts in $\left|\Psi_{0,0}^{\{\mathbf{k}\}}\right\rangle$ Eq. 65, we define the eta-pairing states of the $A$ and $B$ sides:

$$|n_1, n_2, \{m_A\}\rangle = (\hat{\eta}_A^{\dagger})^{n_1}(\hat{\zeta}_A^{\dagger})^{n_2}|\{m_A\}\rangle \tag{69}$$

Using the same steps as in Eq. 57, it is easy to prove orthonormality of $|n_1, n_2, \{m_A\}\rangle$:

$$
\begin{aligned}
\langle n_1', n_2', \{m_A'\}|n_1, n_2, \{m_A\}\rangle &= \frac{1}{n_1! n_2! n_1'! n_2'!} \oint \oint \oint \oint \frac{dz_3}{2\pi i}\frac{dz_4}{2\pi i}\frac{dz_2}{2\pi i}\frac{dz_1}{2\pi i}\frac{1}{z_3^{n_1+1}}\frac{1}{z_4^{n_2+1}}\frac{1}{z_2^{n_1'+1}}\frac{1}{z_1^{n_2'+1}} \\
&\times \Big\langle\{m_A'\}\Big|\prod_{\mathbf{r}\in A}(1 + z_1 \hat{a}_r \hat{c}_{\mathbf{r}\downarrow}^{\dagger})\prod_{\mathbf{r}'\in A}(1 + z_2 e^{-i\pi\cdot\mathbf{r}'}\hat{a}_{r'}\hat{c}_{\mathbf{r}'\downarrow}) \\
&\times \prod_{\mathbf{r}''\in A}(1 + z_3 e^{i\pi\cdot\mathbf{r}''}\hat{c}_{\mathbf{r}'',\downarrow}^{\dagger}\hat{a}_{r''}^{\dagger})\prod_{\mathbf{r}'''\in A}(1 + z_4 \hat{b}_{r'''}\hat{a}_{r'''}^{\dagger})\Big|\{m_A'\}\Big\rangle \\
&= \frac{1}{n_1! n_2! n_1'! n_2'!} \oint \oint \oint \oint \frac{dz_3}{2\pi i}\frac{dz_4}{2\pi i}\frac{dz_2}{2\pi i}\frac{dz_1}{2\pi i}\frac{1}{z_3^{n_1+1}}\frac{1}{z_4^{n_2+1}}\frac{1}{z_2^{n_1'+1}}\frac{1}{z_1^{n_2'+1}} \\
&\times \Big\langle\{m_A'\}\Big|\prod_{\mathbf{r}\in A}(1 + z_1 z_4 \hat{c}_{\mathbf{r}\downarrow}^{\dagger}\hat{c}_{\mathbf{r}\downarrow} + z_2 z_3 \hat{c}_{\mathbf{r}\downarrow}\hat{c}_{\mathbf{r}\downarrow}^{\dagger})\Big|\{m_A\}\Big\rangle \tag{70}
\end{aligned}
$$

We have followed the same steps as in Eq. 57. The manipulations of the operators inside the expectation value so far do not depend on the states *as long as* the left and right states do not contain any $\uparrow$ particles, which the states $|\{m_A\}\rangle$ satisfy. Using then the identical steps as below Eq. 57 we have:

$$
\begin{aligned}
\langle n_1', n_2', \{m_A'\}|n_1, n_2, \{m_A\}\rangle &= \frac{1}{n_1! n_2! n_1'! n_2'!} \oint \oint \oint \oint \frac{dz_3}{2\pi i}\frac{dz_4}{2\pi i}\frac{dz_2}{2\pi i}\frac{dz_1}{2\pi i}\frac{1}{z_3^{n_1+1}}\frac{1}{z_4^{n_2+1}}\frac{1}{z_2^{n_1'+1}}\frac{1}{z_1^{n_2'+1}} \\
&\times (1 + z_3 z_5)^{M_A - N_A}(1 + z_4 z_6)^{N_A}\delta_{\{m_A'\},\{m_A\}} \\
&= \delta_{\{m_A'\},\{m_A\}}\delta_{n_1,n_1'}\delta_{n_2,n_2'}\binom{M_A - N_A}{n_1}\binom{N_A}{n_2}(n_1!)^2(n_2!)^2 \tag{71}
\end{aligned}
$$

We now can find, using the *normalized* $\left|\Psi_{N_1,N_2}^{\{\mathbf{k}\}}\right\rangle$ we find (the limits in the sum are obvious,

for example in binomial coefficients, etc ):

$$
\mathrm{Tr}_A \left| \Psi_{N_1,N_2}^{\{\mathbf{k}\}} \right\rangle \left\langle \Psi_{N_1,N_2}^{\{\mathbf{k}\}} \right|
$$

$$
= \frac{1}{\binom{M-N_{\mathbf{k}}}{N_1}\binom{N_{\mathbf{k}}}{N_2}} \oint \oint \oint \oint \frac{dz_1}{2\pi i}\frac{dz_2}{2\pi i}\frac{dz_3}{2\pi i}\frac{dz_4}{2\pi i} \frac{1}{z_1^{N_1+1}}\frac{1}{z_2^{N_2+1}}\frac{1}{z_3^{N_1+1}}\frac{1}{z_4^{N_2+1}}
$$

$$
\times \sum_{\{m_A\}} \alpha_{\{m_A\}}^2 \sum_{n_1,n_2} \binom{M_A-N_A}{n_1}\binom{N_A}{n_2} (z_1 z_3)^{n_1} (z_2 z_4)^{n_2} \sum_{n_3,n_4,n_5,n_6} \frac{1}{n_3! n_4! n_5! n_6!} z_1^{n_3} z_2^{n_4} z_3^{n_5} z_4^{n_6}
$$

$$
\times (\hat{\eta}_B^{\dagger})^{n_3}(\hat{\zeta}_B^{\dagger})^{n_4} |\{m_B\}\rangle \langle\{m_B\}| (\hat{\zeta}_B)^{n_6}(\hat{\eta}_B)^{n_5}
$$

$$
= \frac{1}{\binom{M-N_{\mathbf{k}}}{N_1}\binom{N_{\mathbf{k}}}{N_2}} \sum_{\{m_A\}} \alpha_{\{m_A\}}^2 \sum_{n_1,n_2} \binom{M_A-N_A}{n_1}\binom{N_A}{n_2} \frac{1}{((N_1-n_1)!)^2}\frac{1}{((N_2-n_2)!)^2}
$$

$$
\times (\hat{\eta}_B^{\dagger})^{N_1-n_1}(\hat{\zeta}_B^{\dagger})^{N_2-n_2} |\{m_B\}\rangle \langle\{m_B\}| (\hat{\zeta}_B)^{N_2-n_2}(\hat{\eta}_B)^{N_1-n_1}
$$

$$
= \frac{1}{\binom{M-N_{\mathbf{k}}}{N_1}\binom{N_{\mathbf{k}}}{N_2}} \sum_{\{m_A\}} \alpha_{\{m_A\}}^2 \sum_{n_1,n_2} \binom{M_A-N_A}{n_1}\binom{N_A}{n_2}\binom{M_A-(N_{\mathbf{k}}-N_A)}{N_1-n_1}
$$

$$
\times \binom{N_{\mathbf{k}}-N_A}{N_2-n_2} |N_1-n_1,N_2-n_2,\{m_B\}\rangle \langle N_1-n_1,N_2-n_2,\{m_B\}| \tag{72}
$$

where $|N_1-n_1,N_2-n_2,\{m_B\}\rangle$ is a normalized state of $N_{\mathbf{k}}-N_A-N_2+n_2+N_1-n_1 \downarrow$ particles and $N_1-n_1+N_2-n_2 \uparrow$ particles on the $B$-side. The limits in the summations over $n_1,n_2$ are implicit from the binomial formulas. The above expression gives the exact entanglement spectrum. By Vandermonde identity, one can check that the trace of the density matrix is unity. With the exact entanglement spectrum, it straightforward to obtain the expression of the Von Neumann entanglement entropy

$$
S_A^{\mathbf{k}} = \sum_{\{m_A\}} \alpha_{\{m_A\}}^2 \log\left(\alpha_{\{m_A\}}^2\right) + \sum_{\{m_A\}} \alpha_{\{m_A\}}^2 \sum_{n_1,n_2} \lambda_{n_1,n_2}^{\mathbf{k}} \log\left(\lambda_{n_1,n_2}^{\mathbf{k}}\right) \tag{73}
$$

where

$$
\lambda_{n_1,n_2}^{\mathbf{k}} = \frac{\binom{M_A-N_A}{n_1}\binom{N_A}{n_2}\binom{M_A-(N_{\mathbf{k}}-N_A)}{N_1-n_1}\binom{N_{\mathbf{k}}-N_A}{N_2-n_2}}{\binom{M-N_{\mathbf{k}}}{N_1}\binom{N_{\mathbf{k}}}{N_2}} \tag{74}
$$

Note that for $N_1 = N$ and $N_2 = 0$, Eqs. 73 and 74 reduce to Eqs. 21 and 19 in the main text.

## D   Thermodynamic limit and scaling

In this section, we give a detailed derivation of the entropy formula in the thermodynamic limit discussed in the main text. For sake of simplicity, we will focus on the case where $N_1 = N$ and $N_2 = 0$.

### D.1   Von Neumann entropy in the thermodynamic limit

The factor $\sum_{\{m_A\}} \alpha_{\{m_A\}}^2$ in Eq. 73 is peaked about $N_A^* \approx \frac{M_A}{M} N_{\mathbf{k}}$ which is large. Therefore, $M_A - N_A$ in $\lambda_{n,0}^{\mathbf{k}}$ is large, and so is $M - N_{\mathbf{k}} - (M_A - N_A)$. This forces the peak of $\lambda_{n,0}^{\mathbf{k}}$ to appear at large $n$. Thus we can use the Stirling's approximation, which gives

$$\lambda_{n,0}^{\mathbf{k}} \approx \frac{1}{\sqrt{2\pi\kappa}} e^{-\frac{1}{2\kappa}(n - n_{\max})^2} \tag{75}$$

$$\kappa = \left(1 - \frac{N}{M - N_{\mathbf{k}}}\right) \frac{N}{M - N_{\mathbf{k}}} \left(1 - \frac{M_A - N_A}{M - N_{\mathbf{k}}}\right)(M_A - N_A) \tag{76}$$

$$\sum_{n=0}^{M_A - N_A} \lambda_{n,0}^{\mathbf{k}} \ln \lambda_{n,0}^{\mathbf{k}} \approx \int_{-\infty}^{\infty} \frac{dn}{\sqrt{2\pi\kappa}} e^{-\frac{1}{2\kappa}(n - n_{\max})^2} \ln\left(\frac{1}{\sqrt{2\pi\kappa}} e^{-\frac{1}{2\kappa}(n - n_{\max})^2}\right) = \tag{77}$$

$$= -\frac{1}{2}\left(1 + \ln(2\pi\kappa)\right) \tag{78}$$

So,

$$S_A^{\mathbf{k}} \approx -\sum_{N_A} \sum_{\{m_A\}} \alpha_{\{m_A\}}^2 \ln \alpha_{\{m_A\}}^2 + \sum_{N_A} \sum_{\{m_A\}} \alpha_{\{m_A\}}^2 \frac{1}{2}\left(1 + \ln(2\pi\kappa)\right)$$

$$\approx -\sum_{N_A} \sum_{\{m_A\}} \alpha_{\{m_A\}}^2 \ln \alpha_{\{m_A\}}^2 + \frac{1}{2}\left(1 + \ln\left(2\pi\kappa_{N_A^*}\right)\right) \tag{79}$$

where

$$\kappa_{N_A^*} = \left(1 - \frac{N}{M - N_{\mathbf{k}}}\right) \frac{N}{M - N_{\mathbf{k}}} \left(1 - \frac{M_A - \frac{M_A}{M} N_{\mathbf{k}}}{M - N_{\mathbf{k}}}\right)(M_A - \frac{M_A}{M} N_{\mathbf{k}}), \tag{80}$$

and using that $\sum_{\{m_A\}} \alpha_{\{m_A\}}^2$ is sharply peaked about $N_A^* \approx \frac{M_A}{M} N_{\mathbf{k}}$ and that $\frac{1}{2}\left(1 + \ln(2\pi\kappa)\right)$ is a smooth function of $N_A$.

Define the density of pairs and the density of $k$'s (magnetization density) as

$$\nu = \frac{N}{M} \qquad \text{and} \qquad \nu_{\mathbf{k}} = \frac{N_{\mathbf{k}}}{M} \tag{81}$$

then

$$\kappa_{N_A^*} = \nu\left(1 - \frac{\nu}{1 - \nu_{\mathbf{k}}}\right)\left(1 - \frac{M_A}{M}\right)M_A, \tag{82}$$

which gives

$$S_A^{\mathbf{k}} \approx -\sum_{N_A} \sum_{\{m_A\}} \alpha_{\{m_A\}}^2 \ln \alpha_{\{m_A\}}^2 + \frac{1}{2}\left(1 + \ln\left(2\pi\nu\left(1 - \frac{\nu}{1 - \nu_{\mathbf{k}}}\right)\left(1 - \frac{M_A}{M}\right)M_A\right)\right) \tag{83}$$

as given in the main text.

### D.2   On why there must be a single peak in $\sum_{\{m_A\}} \alpha_{\{m_A\}}^2$ as a function $N_A$

Start from the saddle point equations (without the Gaussian correction, this does not change the existence of the peak):

$$\sum_{\{m_A\}} \alpha_{\{m_A\}}^2 \approx e^{-(N_A + 1)\ln z_0 + \sum_{m=1}^{N_{\mathbf{k}}} \ln(1 - \gamma_m + z_0 \gamma_m)} = e^{\Phi(N_A)} \tag{84}$$

where $z_0$ is defined by the implicit equation

$$N_A + 1 = z_0 \sum_{m=1}^{N_{\mathbf{k}}} \frac{\gamma_m}{(1 - \gamma_m) + z_0 \gamma_m} \tag{85}$$

Note that $\Phi(N_A)$ depends on $N_A$ both explicitly AND implicitly through the dependence of $z_0$ on $N_A$.

Then,

$$\frac{d\Phi(N_A)}{dN_A} = -\ln z_0 - (N_A + 1)\frac{1}{z_0}\frac{dz_0}{dN_A} + \sum_{m=1}^{N_{\mathbf{k}}} \frac{\gamma_m}{1 - \gamma_m + z_0 \gamma_m}\frac{dz_0}{dN_A} \tag{86}$$

$$= -\ln z_0 \tag{87}$$

because the last two terms cancel due to the saddle point equation. Therefore, the extrema occur when $z_0 = 1$.

If we understand the dependence of $z_0$ on $N_A$, we understand how many extrema there are in $\Phi$ and therefore in $\sum_{\{m_A\}} \alpha_{\{m_A\}}^2$. But, we will now show that $z_0(N_A)$ is a monotonically increasing function of its argument. First, note that we can solve the saddle point equation by taking $z_0 \to 0$, which makes the right-hand-side vanish, therefore $z_0 = 0$ is the solution for $N_A = -1$. Similarly, for $z_0 \to \infty$, the right-hand-side gives $N_{\mathbf{k}}$, therefore, $z_0 \to \infty$ for $N_A = N_{\mathbf{k}} - 1$. Now we have the two limiting cases: $z_0(-1) = 0$ and $z_0(N_{\mathbf{k}} - 1) \to \infty$, and at these two points $\frac{d\Phi(N_A)}{dN_A} > 0$ and $\frac{d\Phi(N_A)}{dN_A} < 0$, respectively.

Now, take the derivative of both sides of the saddle point equation with respect to $N_A$. We get,

$$1 = \sum_{m=1}^{N_{\mathbf{k}}} \frac{\gamma_m(1 - \gamma_m)}{\left(\frac{1}{z_0}(1 - \gamma_m) + \gamma_m\right)^2}\frac{1}{z_0^2}\frac{dz_0}{dN_A} \tag{88}$$

leading to

$$\frac{dz_0}{dN_A} = \frac{z_0^2}{\sum_{m=1}^{N_{\mathbf{k}}} \frac{\gamma_m(1 - \gamma_m)}{\left(\frac{1}{z_0}(1 - \gamma_m) + \gamma_m\right)^2}} \geq 0 \tag{89}$$

because $0 < \gamma_m < 1$. This proves that $z_0(N_A)$ is monotonically increasing from 0 to $\infty$ as $N_A$ goes from $-1$ to $N_{\mathbf{k}} - 1$. Therefore, there is a single value of $N_A$ at which $z_0 = 1$, which is where $\frac{d\Phi(N_A)}{dN_A} = 0$. Since $\frac{d\Phi(N_A)}{dN_A}\big|_{N_A = -1} > 0$ and $\frac{d\Phi(N_A)}{dN_A}\big|_{N_A = N_{\mathbf{k}} - 1} < 0$, the value $N_A$ at which $z_0 = 1$ is the maximum of $\Phi$.

How sharp is the maximum? Denote the value of $N_A$ which maximizes $\Phi$ by $N_A^*$. As shown above,

$$z_0(N_A^*) = 1. \tag{90}$$

Thus,

$$\sum_{\{m_A\}} \alpha_{\{m_A\}}^2 \approx e^{\Phi(N_A^*)} \exp\left(\frac{(N_A - N_A^*)^2}{2}\frac{d^2\Phi}{dN_A^{*2}}\right) \tag{91}$$

But,

$$\frac{d^2\Phi(N_A)}{dN_A^2} = -\frac{1}{z_0}\frac{dz_0}{dN_A} = -\frac{z_0}{\sum_{m=1}^{N_{\mathbf{k}}} \frac{\gamma_m(1 - \gamma_m)}{\left(\frac{1}{z_0}(1 - \gamma_m) + \gamma_m\right)^2}}. \tag{92}$$

Evaluating this at $N_A^*$ gives

$$0 > -\frac{d^2\Phi}{dN_A^{*2}} = \frac{1}{\sum_{m=1}^{N_{\mathbf{k}}} \gamma_m(1-\gamma_m)} = \frac{1}{\mathrm{Tr}\Gamma - \mathrm{Tr}\Gamma^2}$$

$$\approx \frac{M}{M_A N_{\mathbf{k}}\left(1 - \frac{M_A}{M} - \frac{N_{\mathbf{k}}}{M}\right)} \approx \mathcal{O}\left(\frac{1}{N_{\mathbf{k}}}\right). \tag{93}$$

where

$$\Gamma_{\mathbf{kk'}} = \frac{1}{M}\sum_{\mathbf{r}\in A} e^{-i(\mathbf{k}-\mathbf{k'})\cdot\mathbf{r}} \tag{94}$$

is the one-body density matrix already introduced in Eq. 58. Therefore,

$$\sum_{\{m_A\}}\alpha_{\{m_A\}}^2 \approx e^{\Phi(N_A^*)}\exp\left(-\frac{(N_A-N_A^*)^2}{2\rho N_{\mathbf{k}}}\right), \quad \rho \sim \mathcal{O}(1) \tag{95}$$

The width of the peak is therefore of order $\sqrt{N_{\mathbf{k}}}$. Therefore, as long as $N_A^* \gg 1$, the peak is sharp.

But we can actually determine the value of $N_A^*$. Indeed going back to the saddle point equation, we must have

$$N_A^* + 1 = \sum_{m=1}^{N_{\mathbf{k}}} \gamma_m = \mathrm{Tr}\,\Gamma \tag{96}$$

Since the trace of the one-body density matrix satisfies

$$\mathrm{Tr}\,\Gamma = \sum_{\mathbf{k}}\Gamma_{\mathbf{kk}} = \frac{M_A}{M}N_{\mathbf{k}} \tag{97}$$

we get,

$$N_A^* = \frac{M_A}{M}N_{\mathbf{k}}. \tag{98}$$

leading in the thermodynamic limit to the formula

$$\sum_{\{m_A\}}\alpha_{\{m_A\}}^2 \to \frac{1}{\sqrt{2\pi(\mathrm{Tr}\,\Gamma - \mathrm{Tr}\,\Gamma^2)}}\exp\left(-\frac{(N_A-\mathrm{Tr}\,\Gamma)^2}{2(\mathrm{Tr}\,\Gamma - \mathrm{Tr}\,\Gamma^2)}\right) \tag{99}$$

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
