# Peer review of "Entanglement of Exact Excited Eigenstates of the Hubbard Model in Arbitrary Dimension"

_SciPost Physics, doi:SciPost Phys. 3, 043 (2017)_

## Round 2 · Referee Report · Anonymous (Referee 1) · 2017-10-31

Strengths
(2) elegant analysis
(3) Very clear exposition and discussion of results
Weaknesses
Report
Requested changes
(1) The authors may want to consider citing the work by Doyon and Castro-Alvaredo Phys. Rev. Lett. 108, 120401 (2012)
(2) In the thermal density matrix specified below (24) I assume the authors mean $\hat{J}^2$ rather than $\hat{J}$

---

## Round 2 · Referee Report · Anonymous (Referee 2) · 2017-11-21

Strengths
-
Elegant and clean computation of the entanglement of certain highly excited eigenstates of an interacting fermion system.
-
Allows a controlled study of the validity of the eigenstate thermalization hypothesis.
Weaknesses
-
The authors' results concerning the entanglement are perhaps not (a posteriori) that surprising given that the eigenstates amount to free fermion states dressed with a "condensate". The most nontrivial thing is the very existence of these eigenstates, pointed out by Yang almost 30 years ago.
-
Conclusions as pertaining to eigenstate thermalization are not clearly stated.
Report
It is not completely clear to me what general lessons are learnt from the computations in this paper. Nonetheless, because tractable, highly excited exact eigenstates of interacting models are few and far between, it is certainly worth studying these states. The results are cleanly presented and I believe will be useful in future attempts to sharpen the eigenstate thermalization hypothesis and possibly in future studies of the Hubbard model. I found the contour integral computation of the entanglement in the eta-paired states very elegant, although I do not know whether this method is novel or not.
I recommend publishing this paper after consideration of the minor changes suggested below.
Requested changes
I have only minor suggestions for changes to improve readability.
(1) Much of the supplementary material is concerned with a "generalized" Hubbard model, yet this model is not mentioned at any point in the main text (except in a footnote). If the authors actually care about this generalization, perhaps they want to at least mention it somewhere in the main text.
(2) The paper is motivated by the desire to examine the validity of ETH, yet in the abstract, introduction and conclusion there is not a clean statement to be found about the take-away message from the paper in this regard. In the middle of the text there is a discussion about the fact that the states considered are closely related to the representation theory of a pseudospin symmetry, and this theme is picked up again in a paragraph near the end that compares to a thermal density matrix. I would recommend that the authors add some sharp statements in some or all of the abstract, introduction and conclusion that would allow a reader that is browsing the text quickly to see what the take-home messages are regarding ETH.
(3) The final sentence in the abstract talks about the onset of thermalization. This question is then not mentioned anywhere in the main text. This sentence then probably belongs in the conclusion rather than the abstract.
(4) Below equation (3) it would be useful to say what the vector pi is.
(5) Below equation (3) it would be useful to very briefly (one or two sentences) outline why these are eigenstates, just to make the reading more fluid.
(6) Somewhere below equation (11) it would be helpful for the reader to state what fraction of the total number of states can be written in this form. I.e. a measure of how "typical" or not they are. It would also be useful to explicitly clarify whether the N and N_k spins can have overlap or need to be chosen from disjointed sets.

---

## Editorial Decision

published